# MoCo-CXR: MoCo Pretraining Improves Representation and Transferability of Chest X-ray Models

**Hari Sowrirajan**[*]                  HSOWRIRA@STANFORD.EDU
**Jingbo Yang**[*]                    JINGBOY@STANFORD.EDU
**Andrew Y. Ng**                    ANG@CS.STANFORD.EDU
**Pranav Rajpurkar**                PRANAVSR@CS.STANFORD.EDU
*Department of Computer Science, Stanford University*

## Abstract

Contrastive learning is a form of self-supervision that can leverage unlabeled data to produce pretrained models. While contrastive learning has demonstrated promising results on natural image classification tasks, its application to medical imaging tasks like chest X-ray interpretation has been limited. In this work, we propose MoCo-CXR, which is an adaptation of the contrastive learning method Momentum Contrast (MoCo), to produce models with better representations and initializations for the detection of pathologies in chest X-rays. In detecting pleural effusion, we find that linear models trained on MoCo-CXR-pretrained representations outperform those without MoCo-CXR-pretrained representations, indicating that MoCo-CXR-pretrained representations are of higher-quality. End-to-end fine-tuning experiments reveal that a model initialized via MoCo-CXR-pretraining outperforms its non-MoCo-CXR-pretrained counterpart. We find that MoCo-CXR-pretraining provides the most benefit with limited labeled training data. Finally, we demonstrate similar results on a target Tuberculosis dataset unseen during pretraining, indicating that MoCo-CXR-pretraining endows models with representations and transferability that can be applied across chest X-ray datasets and tasks.

**Keywords:** Contrastive Learning, Chest X-Rays

## 1. Introduction

Self-supervised approaches such as Momentum Contrast (MoCo) (He et al., 2019; Chen et al., 2020c) can leverage unlabeled data to produce pretrained models for subsequent fine-tuning on labeled data. Contrastive learning of visual representations has emerged as the front-runner for self-supervision and has demonstrated superior performance on downstream tasks. In addition to MoCo, these include frameworks such as SimCLR (Chen et al., 2020a,b) and PIRL (Misra and Maaten, 2020). All contrastive learning frameworks involve maximizing agreement between positive image pairs relative to negative/different images via a contrastive loss function; this pretraining paradigm forces the model to learn good representations. These approaches typically differ in how they generate positive and negative image pairs from unlabeled data and how the data are sampled during pretraining. While MoCo and other contrastive learning methods have demonstrated promising results on natural image classification tasks, their application to medical imaging settings has been limited (Raghu et al., 2019; Cheplygina et al., 2019).

Chest X-ray is the most common imaging tool used in practice, and is critical for screening, diagnosis, and management of diseases. The recent introduction of large datasets (size 100k-500k)

---

[*] Contributed equally

of chest X-rays (Irvin et al., 2019; Johnson et al., 2019; Bustos et al., 2020) has driven the development of deep learning models that can detect diseases at a level comparable to radiologists (Rajpurkar et al., 2020, 2021). Because there is an abundance of unlabeled chest X-ray data (Raoof et al., 2012), contrastive learning approaches represent a promising avenue for improving chest X-ray interpretation models.

Chest X-ray interpretation is fundamentally different from natural image classification in that (1) disease classification may depend on abnormalities in a small number of pixels, (2) data attributes for chest X-rays differ from natural image classification because X-rays are larger, grayscale and have similar spatial structures across images (always either anterior-posterior, posterior-anterior, or lateral), (3) there are far fewer unlabeled chest X-ray images than natural images. These differences may limit the applicability of contrastive learning approaches, which were developed for natural image classification settings, to chest X-ray interpretation. For instance, MoCo uses a variety of data augmentations to generate positive image pairs from unlabeled data; however, random crops and blurring may eliminate disease-covering parts from an augmented image, while color jittering and random gray scale would not produce meaningful transformations for already grayscale images. Furthermore, given the availability of orders of magnitude fewer chest X-ray images than natural images, and larger image sizes, it remains to be understood whether retraining may improve on the traditional paradigm for automated chest X-ray interpretation, in which models are fine-tuned on labeled chest X-ray images from ImageNet-pretrained weights.

In this work, we demonstrate that our proposed MoCo-CXR method leads to better representations and initializations than those acquired without MoCo-pretraining (solely from ImageNet) for chest X-ray interpretation. The MoCo-CXR pipeline involved first a modified MoCo-pretraining on CheXpert (Irvin et al., 2019), where we adapted initialization, data augmentations, and learning rate scheduling of this pretraining step for successful application on chest X-rays. This was then followed by supervised fine-tuning experiments using different fractions of labeled data. We showed that MoCo-CXR-pretrained representations are of higher quality than ImageNet-pretrained representations by evaluating the performance of a linear classifier trained on pretrained representations on a chest X-ray interpretation task. We also demonstrated that a model trained end-to-end with MoCo-CXR-pretrained initialization had superior performance on the X-ray interpretation tasks, and the advantage was especially apparent at low labeled data regimes. Finally, we also showed that MoCo-CXR-pretrained representations from the source (CheXpert) dataset transferred to another small chest X-ray dataset (Shenzhen) with a different classification task (Jaeger et al., 2014). Our study demonstrates that MoCo-CXR provides high-quality representations and transferable initializations for chest X-ray interpretation.

## 2. Related Work

**Self-supervised learning**   Self-supervision is a form of unsupervised pretraining that uses a formulated pretext task on unlabeled data as the training goal. Handcrafted pretext tasks include solving jigsaw puzzles (Noroozi and Favaro, 2016), relative patch prediction (Doersch et al., 2015) and colorization (Zhang et al., 2016). However, many of these tasks rely on ad-hoc heuristics that could limit the generalization and transferability of learned representations for downstream tasks. Consequently, contrastive learning of visual representations has emerged as the front-runner for self-supervision and has demonstrated superior performance on downstream tasks (Chen et al., 2020c,b).

**Contrastive learning for chest X-rays**   Prior work using contrastive learning on chest X-rays is limited in its applicability to unlabeled data and evidence of transferability. A controlled approach is to explicitly contrast X-rays with pathologies against healthy ones using attention network (Liu et al., 2019); however, this approach is supervised. There has also been a proposed domain-specific strategy of extracting contrastive pairs from MRI and CT datasets using a combination of localized contrastive loss function and global loss function during pretraining (Chaitanya et al., 2020). However, the method is highly dependent on the volumetric nature of MRI and CT scans, as the extraction of similar image pairs is based on taking 2D image slices of a single volumetric image. Thus, the technique would have limited applicability to chest X-rays. Work applying broader self-supervised techniques to medical imaging is more extensive. For example, encoding shared information between different imaging modalities for ophthal-

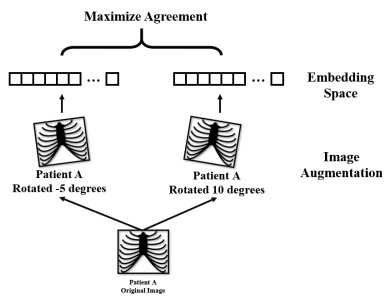

Figure 1:   Contrastive learning maximizes agreement of embeddings generated by different augmentations of the same chest X-ray image.

mology was shown to be an effective pretext task for pretraining diabetic retinopathy classification models (Holmberg et al., 2019).   Other proposed pretext tasks in medical imaging include solving a Rubik's cube (Zhuang et al., 2019; Zhu et al., 2020), predicting the position of anatomical patches (Bai et al., 2019), anatomical reconstruction (Zhou et al., 2019), and image patch distance estimation (Spitzer et al., 2018).

**ImageNet transfer for chest X-ray interpretation**   The dominant computer vision approach of starting with an ImageNet-pretrained model has been proven to be highly effective at improving model performance in diverse settings such as object detection and image segmentation (Holmberg et al., 2019).   Although high performance deep learning models for chest X-ray interpretation use ImageNet-pretrained weights, Sun et al. (2019) found that common regularization techniques limit ImageNet transfer learning benefits and that ImageNet features are less general than previously believed.   Moreover, Zhuang et al. (2019) showed that randomly-initialized models are competitive with their ImageNet-initialized counterparts on a vast array of tasks with sufficient labeled data, and that pretraining merely speeds up convergence.   Raghu et al. (2019) further investigated the efficacy of ImageNet pretraining, observing that simple convolutional architectures are able to achieve comparable performance as larger ImageNet model architectures.

## 3. Methods

### 3.1. Chest X-ray datasets and diagnostic tasks

We used a large source chest X-ray dataset for pretraining and a smaller external chest X-ray dataset for the evaluation of model transferability. The source chest X-ray dataset we select is CheXpert, a large collection of chest X-ray images labeled for the presence or absence of several diseases (Irvin et al., 2019). CheXpert consists of 224k chest X-rays collected from 65k patients. Chest X-ray images included in the CheXpert dataset are of size $320 \times 320$. We focused on identifying the presence of pleural effusion, a clinically important condition that has high prevalence in the dataset (with 45.63% of all images labeled as positive or uncertain). We performed follow-up experiments

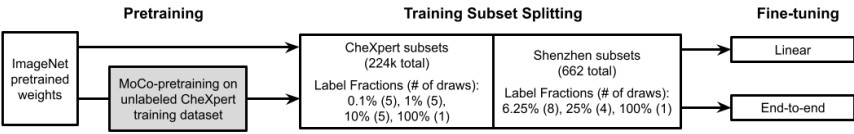

Figure 2: MoCo-CXR training pipeline. MoCo acts as self-supervised training agent. The model is subsequently tuned using chest X-ray images.

with other CheXpert competition tasks (cardiomegaly, consolidation, edema and atelectasis) from Irvin et al. (2019) to verify that our method worked on different pathologies. In addition, we use the Shenzhen Hospital X-ray set (Jaeger et al., 2014) for evaluation of model transferability to an external target dataset. Chest X-ray images included in the Shenzhen dataset are of size $4020 \times 4892$ and $4892 \times 4020$. The Shenzhen dataset contains 662 X-ray images, of which 336 (50.8%) are abnormal X-rays that have manifestations of tuberculosis. All images in both datasets are resized to $224 \times 224$ for MoCo-CXR.

### 3.2. MoCo-CXR Pretraining for Chest X-ray Interpretation

We adapt the MoCo-pretraining procedure to chest X-rays. MoCo is a form of self-supervision that utilizes contrastive learning, where the pretext task is to maximize agreement between different views of the same image (positive pairs) and to minimize agreement between different images (negative pairs). Figure 1 illustrates how data augmentations are used to generate views of a particular image and are subsequently contrasted to learn embeddings in an unsupervised fashion.

Our choice to use MoCo is driven by two constraints in medical imaging AI: (1) large image sizes, and (2) the cost of large computational resources. Compared to other self-supervised frameworks such as SimCLR (Chen et al., 2020a), MoCo requires far smaller batch sizes during pretraining (Chen et al., 2020c). The MoCo implementation used a batch size of 256 and achieved comparable performance on ImageNet as the SimCLR implementation, which used a batch size of 4096; in contrast, SimCLR experienced lower performance at a batch size of 256 (Chen et al., 2020c). MoCo's reduced dependency on mini-batch size is achieved by using a momentum updated queue of previously seen samples to generate contrastive pair encodings. An illustration of MoCo's momentum encoding framework has been added as Appendix Figure 1. Using MoCo, we were able to conduct experiments on a single NVIDIA GTX 1070 with a batch size of 16.

We performed MoCo-pretraining on the entire CheXpert training dataset. We chose to apply MoCo-pretraining on ImageNet-initialized models to leverage possible convergence benefits (Raghu et al., 2019). Due to the widespread availability of ImageNet-pretrained weights, there is no extra cost to initialize models with ImageNet weights before MoCo-pretraining.

We modified the data augmentation strategy used to generate views suitable for the chest X-ray interpretation task. Data augmentations used in self-supervised approaches for natural images may not be appropriate for chest X-rays. For example, random crop and Gaussian blur could change the disease label for an X-ray image or make it impossible to distinguish between diseases. Furthermore, color jittering and random grayscale do not represent meaningful augmentations for grayscale X-rays. Instead, we use random rotation (10 degrees) and horizontal flipping (Appendix Figure 2), a set of augmentations commonly used in training chest X-ray models (Irvin et al., 2019; Rajpurkar et al., 2017) driven by experimental findings in the supervised setting and clinical domain knowl-

edge. Future work should investigate the impact of various additional augmentations and their combinations.

The overall training pipeline with MoCo-CXR-pretraining and the subsequent fine-tuning with CheXpert and Shenzhen datasets is illustrated in Figure 2. We maintained hyperparameters related to momentum, weight decay, and feature dimension from MoCo (Chen et al., 2020c). Checkpoints from top performing epochs were saved for subsequent checkpoint selection and model evaluation. In the subsequent fine-tuning step, we selected hyperparameters based on performance of linear evaluations. We used two backbones, ResNet18 and DenseNet121, to evaluate the consistency of our findings across model architectures. We experimented with initial learning rates of $10^{-2}$, $10^{-3}$, $10^{-4}$ and $10^{-5}$, and investigated their effect on performance. We also experimented with milestone and cosine learning rate schedulers.

### 3.3. MoCo-CXR Model Fine-tuning

We fine-tuned models on different fractions of labeled training data. We also conducted baseline fine-tuning experiments with ImageNet-pretrained models that were not subjected to MoCo-CXR-pretraining. We use label fraction to represent the ratio of data with its labels retained during training. For a model trained with 1% label fraction, the model will only have access to 1% of the all labels, while the remaining 99% of labels are hidden from the model. The use of label fraction is a proxy for the real world, where large amounts of data remain unlabelled and only a small portion of well-labelled data can be used toward supervised training. As presented in Figure 2, the label fractions of training sets are 0.1%, 1%, 10% and 100% for the CheXpert dataset and 6.25%, 25%, 100% for the external Shenzhen dataset. Fine-tuning experiments on small label fractions are repeated multiple times with different random samples and averaged to guard against anomalous, unrepresentative training splits.

To evaluate the transfer of representations, we froze the backbone model and trained a linear classifier on top using the labeled data (MoCo-CXR/ImageNet-pretrained Linear Models). In addition, we unfreeze all layers and fine-tune the entire model end-to-end using the labeled data to assess transferability on the overall performance (MoCo-CXR/ImageNet-pretrained end-to-end Models). Our models were fine-tuned using the same configurations as fully-supervised models designed for CheXpert (Irvin et al., 2019), which has determined an optimal batch size, learning rate and other hyper-parameters. To be specific, we use a learning rate of $3 \times 10^{-5}$, batch size of 16 and number of epochs that scale with the size of labeled data. For the CheXpert dataset, these are 220, 95, 41, 18 epochs for the 4 label fractions respectively.

### 3.4. Statistical analysis

We compared the performance of the models trained with and without MoCo-CXR-pretraining using the area under the receiver operating characteristic curve (AUC). To assess whether MoCo-CXR-pretraining significantly changed the performance, we computed the difference in AUC on the test set with and without MoCo-CXR-pretraining. The non-parametric bootstrap was used to estimate the variability around model performance. A total of 500 bootstrap replicates from the test set were drawn, and the AUC and its corresponding differences were calculated for the MoCo-CXR-pretrained model and non-Moco-CXR pretrained model on these same 500 bootstrap replicates. This produced a distribution for each estimate, and the 95% bootstrap percentile intervals were computed to assess significance at the $p = 0.05$ level.

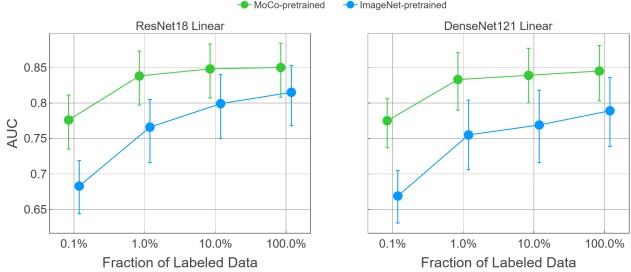

Figure 3: AUC on pleural effusion task for linear models with MoCo-CXR-pretraining is consistently higher than AUC of linear models with ImageNet-pretraining, showing that MoCo-CXR-pretraining produces higher quality representations than ImageNet-pretraining does.

## 4. Experiments

### 4.1. Transfer performance of MoCo-CXR-pretrained representations on CheXpert

We investigated whether representations acquired from MoCo-CXR-pretraining are of higher quality than those transferred from ImageNet. To evaluate the representations, we used the linear evaluation protocol (Oord et al., 2018; Zhang et al., 2016; Kornblith et al., 2019; Bachman et al., 2019), where a linear classifier is trained on a frozen base model, and test performance is used as a proxy for representation quality. We visualize the performance of MoCo-CXR-pretrained and ImageNet-pretrained linear models at various label fractions in Figure 3 and tabulate the corresponding AUC improvements in Table 1.

Trained on small label fractions, the ResNet18 MoCo-CXR-pretrained linear model demonstrated statistically significant performance gains over its ImageNet counterpart. With 0.1% label fraction, the improvement in performance is 0.096 (95% CI 0.061, 0.130) AUC; the MoCo-CXR-pretrained and ImageNet-pretrained linear models achieved performances of 0.776 and 0.683 AUC respectively. These findings support the hypothesis that MoCo-representations are of superior quality, and are most apparent when labeled data is scarce.

With larger label fractions, the MoCo-CXR-pretrained linear models demonstrated clear yet diminishing improvements over the ImageNet-pretrained linear models. Training with 100% of the labeled data, the ResNet18 MoCo-CXR-pretrained linear model yielded a performance gain of 0.034 (95% CI -0.009, 0.080). Both backbones were observed to have monotonically decreasing performance gains with MoCo as we increase the amount of labeled training data. These results provide evidence that MoCo-CXR-pretrained representations retain their quality at all label fractions, but less significantly at larger label fractions. We generally observe similar performance gains with MoCo-CXR on the CheXpert competition pathologies, as seen in Appendix Table 4.

### 4.2. Transfer performance of end-to-end MoCo-CXR-pretrained models on CheXpert

We investigated whether MoCo-CXR-pretraining translates to higher performance for models fine-tuned end-to-end. We visualize the performance of the MoCo and ImageNet-pretrained end-to-end models at different label fractions in Figure 4. AUC improvements of using a MoCo-CXR-pretrained linear model over an ImageNet-pretrained linear is tabulated in Table 1

We found that MoCo-CXR-pretrained end-to-end models outperform their ImageNet-pretrained counterparts more at small label fractions than at larger label fractions. With the 0.1% label frac-

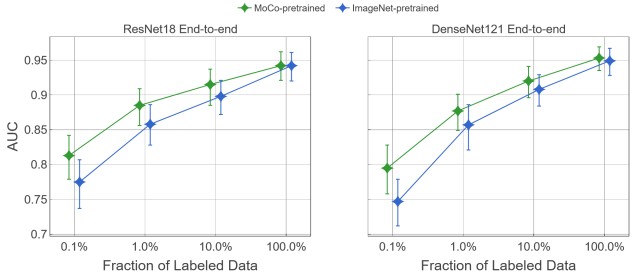

Figure 4: AUC on pleural effusion task for models fine-tuned end-to-end with MoCo-CXR-pretraining is consistently higher than those without MoCo-CXR-pretraining, showing that MoCo-CXR-pretraining representations are more transferable than those produced by ImageNet-pretraining only.

| Architecture | MoCo-CXR-pretrained | ImageNet-pretrained | 0.1% | 1.0% | 10.0% | 100% |
|---|---|---|---|---|---|---|
| ResNet18 | End-to-End | End-to-End | 0.037( 0.015, 0.062) | 0.027( 0.006, 0.047) | 0.017( 0.003, 0.031) | 0.000(-0.009, 0.009) |
| ResNet18 | Linear Model | Linear Model | 0.096( 0.061, 0.130) | 0.070( 0.029, 0.112) | 0.049( 0.005, 0.094) | 0.034(-0.009, 0.080) |
| ResNet18 | Linear Model | End-to-End | 0.001(-0.024, 0.025) | -0.022(-0.051, 0.009) | -0.050(-0.083, -0.018) | -0.094(-0.127, -0.062) |
| DenseNet121 | End-to-End | End-to-End | 0.048( 0.023, 0.074) | 0.019( 0.001, 0.037) | 0.012( 0.000, 0.023) | 0.003(-0.006, 0.013) |
| DenseNet121 | Linear Model | Linear Model | 0.107( 0.075, 0.142) | 0.078( 0.035, 0.121) | 0.067( 0.023, 0.111) | 0.055( 0.008, 0.102) |
| DenseNet121 | Linear Model | End-to-End | 0.029( 0.002, 0.055) | -0.024(-0.050, -0.003) | -0.070(-0.109, -0.036) | -0.107(-0.141, -0.073) |

Table 1: AUC improvements on pleural effusion task achieved by MoCo-CXR-pretrained models against models without MoCo-CXR-pretraining on the CheXpert dataset.

tion, the ResNet18 MoCo-CXR-pretrained end-to-end model achieved an AUC of 0.813 while the ImageNet-pretrained end-to-end model achieves an AUC of 0.775, yielding a statistically significant AUC improvement of 0.037 (95% CI 0.015, 0.062). The AUC improvement with the 1.0% label fraction was also statistically significant at 0.027 (95% CI 0.006, 0.047). Both pretraining approaches converge to an AUC of 0.942 with the 100% label fraction.

These results demonstrate that MoCo-CXR-pretraining yields performance boosts for end-to-end training, and further substantiate the quality of the pretrained initialization, especially for smaller label fractions. This finding is consistent with behavior of SimCLR (Chen et al., 2020a), which also saw larger performance gains for self-supervised models trained end-to-end on smaller label fractions of ImageNet. We generally observe similar performance gains with MoCo-CXR end-to-end on the CheXpert competition pathologies, as seen in Appendix Table 5.

### 4.3. Transfer benefit of MoCo-CXR-pretraining on an external dataset

We conducted experiments to test whether MoCo-CXR-pretrained chest X-ray representations acquired from a source dataset (CheXpert) transfer to a small target dataset (Shenzhen Dataset for Tuberculosis, with 662 X-rays). Results of these experiments are presented in Figure 5.

We first examined whether MoCo-CXR-pretrained linear models improve AUC on the external Shenzhen dataset. With 6.25% label fraction, which is approximately 25 images, the ResNet18 MoCo-CXR-pretrained model outperformed the ImageNet-pretrained one by 0.054 (95%, CI 0.024, 0.086). AUC improvement with the 100% label fraction was 0.018 (95% CI -0.011, 0.053). This is similar to the trend observed on the CheXpert dataset discussed previously. These observations sug-

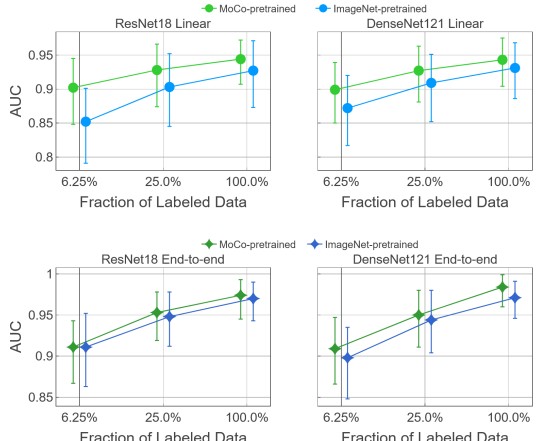

| Learning Rate | AUC |
|:---:|:---:|
| $10^{-2}$ | 0.786 (0.699, 0.861) |
| $10^{-3}$ | 0.908 (0.853, 0.955) |
| $10^{-4}$ | 0.944 (0.907, 0.972) |
| $10^{-5}$ | 0.939 (0.891, 0.975) |

Figure 5: AUC on the Shenzhen tuberculosis task for models with and without MoCo-CXR-pretraining shows that MoCo pretraining still introduces significant improvement despite being fine-tuned on an external dataset.

Table 2: AUC of MoCo pretrained ResNet18 on Shenzhen dataset at different pretraining learning rates with $100\%$ label fraction.

gest that representations learned from MoCo-CXR-pretraining are better suited for an external target chest X-ray dataset with a different task than representations learned from ImageNet-pretraining.

Next, we tested whether MoCo-CXR-pretrained models with end-to-end training also perform well on the external Shenzhen dataset. With the 100% label fraction, the ResNet18 MoCo-CXR-pretrained model was able to achieve an AUC of 0.974. However, the corresponding AUC improvement of only 0.003 (95% CI -0.014, 0.020) is much less than the improvement observed for linear models. Since the Shenzhen dataset is limited in size, it is possible that training end-to-end quickly saturates learning potential at low label fractions. Regardless, the non-zero improvement still suggests that MoCo-CXR-pretrained initializations can transfer to an external dataset. This echoes previous investigations of self-supervised and unsupervised learnings, which found that unsupervised pretraining pushes the model towards solutions with better generalization to tasks that are in the same domain (Sun et al., 2019; Erhan et al., 2010).

## 5. Conclusion

We found that our MoCo-CXR method provides high-quality representations and transferable initializations for chest X-ray interpretation. Despite many differences in the data and task properties between natural image classification and chest X-ray interpretation, MoCo-CXR was a successful adaptation of MoCo pretraining to chest X-rays. These suggest the possibility for broad application of self-supervised approaches beyond natural image classification settings.

To the best of our knowledge, this work is the first to show the benefit of MoCo-pretraining across label fractions for chest X-ray interpretation, and also investigate representation transfer to a target dataset. All of our experiments are run on a single NVIDIA GTX 1070, demonstrating accessibility of this method. Our success in demonstrating improvements in model performance over the traditional supervised learning approach, especially on low label fractions, may be broadly extensible to other medical imaging tasks and modalities, where high-quality labeled data is expensive, but unlabeled data is increasingly easier to access.

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

## Appendix A.  Supplementary Details for the MoCo-CXR Method

Source code is available on Github.

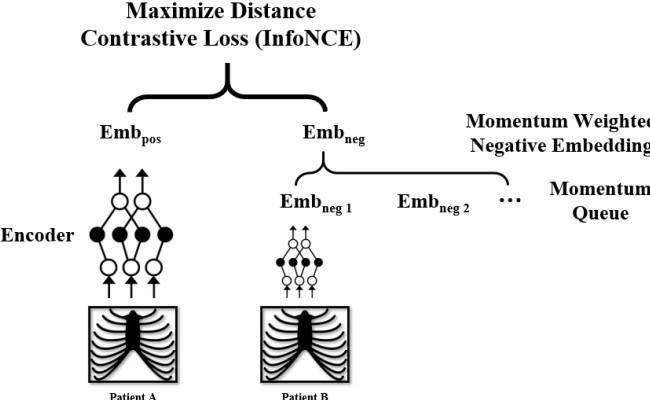

Figure 1: The MoCo framework generates negative embeddings in a momentum-weighted manner using a queue of negative embeddings. This setup reduces dependency on batch size, therefore has more relaxed hardware constraint compared to other self-supervised learning frameworks.

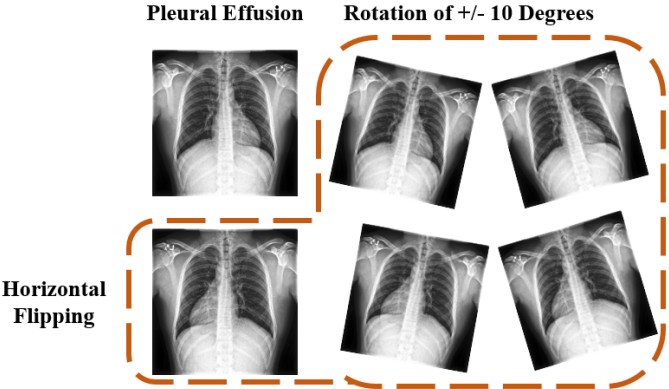

Figure 2: Illustration of data augmentation methods used for MoCo-CXR, which are horizontal flip and random rotations for data augmentation.

## Appendix B.  Supplementary Details for MoCo-CXR Performance

| Pretraining | Architecture | Fine Tuning | 0.1% | 1.0% | 10.0% | 100.0% |
|---|---|---|---|---|---|---|
| MoCo | ResNet18 | Linear Model | 0.776(0.735, 0.811) | 0.838(0.797, 0.873) | 0.848(0.807, 0.883) | 0.850(0.808, 0.884) |
| ImageNet | ResNet18 | Linear Model | 0.683(0.644, 0.719) | 0.766(0.716, 0.805) | 0.799(0.750, 0.840) | 0.815(0.768, 0.853) |
| MoCo | DenseNet121 | Linear Model | 0.775(0.737, 0.806) | 0.833(0.790, 0.871) | 0.839(0.800, 0.877) | 0.845(0.803, 0.881) |
| ImageNet | DenseNet121 | Linear Model | 0.669(0.631, 0.705) | 0.755(0.706, 0.804) | 0.769(0.716, 0.818) | 0.789(0.739, 0.836) |
| MoCo | ResNet18 | End-to-end | 0.813(0.779, 0.842) | 0.885(0.856, 0.909) | 0.915(0.885, 0.937) | 0.942(0.921, 0.962) |
| ImageNet | ResNet18 | End-to-end | 0.775(0.737, 0.807) | 0.858(0.828, 0.886) | 0.898(0.872, 0.921) | 0.942(0.920, 0.961) |
| MoCo | DenseNet121 | End-to-end | 0.795(0.758, 0.828) | 0.877(0.849, 0.901) | 0.920(0.896, 0.941) | 0.953(0.935, 0.969) |
| ImageNet | DenseNet121 | End-to-end | 0.747(0.712, 0.779) | 0.857(0.821, 0.886) | 0.908(0.884, 0.929) | 0.949(0.928, 0.967) |

Table 1: Table corresponding to Main Figure 3 and Figure 4.  AUC of models trained to detect pleural effusion on the CheXpert dataset.

| Pretraining | Architecture | Fine Tuning | 6.25% | 25.0% | 100.0% |
|---|---|---|---|---|---|
| MoCo | ResNet18 | Linear Model | 0.902(0.848, 0.945) | 0.928(0.874, 0.966) | 0.944(0.907, 0.972) |
| ImageNet | ResNet18 | Linear Model | 0.852(0.791, 0.901) | 0.903(0.845, 0.952) | 0.927(0.873, 0.971) |
| MoCo | DenseNet121 | Linear Model | 0.899(0.850, 0.939) | 0.927(0.881, 0.963) | 0.943(0.904, 0.975) |
| ImageNet | DenseNet121 | Linear Model | 0.872(0.817, 0.920) | 0.909(0.852, 0.951) | 0.931(0.886, 0.968) |
| MoCo | ResNet18 | End-to-end | 0.911(0.867, 0.943) | 0.953(0.919, 0.978) | 0.974(0.945, 0.993) |
| ImageNet | ResNet18 | End-to-end | 0.911(0.863, 0.952) | 0.948(0.912, 0.978) | 0.970(0.943, 0.990) |
| MoCo | DenseNet121 | End-to-end | 0.909(0.866, 0.947) | 0.950(0.911, 0.980) | 0.984(0.960, 0.999) |
| ImageNet | DenseNet121 | End-to-end | 0.898(0.848, 0.935) | 0.944(0.904, 0.980) | 0.971(0.946, 0.991) |

Table 2: Table corresponding to Main Figure 5.  AUC of models trained to detect tuberculosis on the Shenzhen dataset.

| Architecture | MoCo-CXR-pretrained | ImageNet-pretrained | 6.25% | 25.0% | 100% |
|---|---|---|---|---|---|
| ResNet18 | End-to-End | End-to-End | 0.001(-0.022, 0.027) | 0.005(-0.012, 0.027) | 0.003(-0.014, 0.020) |
| ResNet18 | Linear Model | Linear Model | 0.054( 0.024, 0.086) | 0.026(-0.001, 0.056) | 0.018(-0.011, 0.053) |
| ResNet18 | Linear Model | End-to-End | -0.007(-0.029, 0.015) | -0.020(-0.040, -0.003) | -0.026(-0.052, -0.005) |
| DenseNet121 | End-to-End | End-to-End | 0.011(-0.006, 0.028) | 0.006(-0.010, 0.023) | 0.013(-0.003, 0.033) |
| DenseNet121 | Linear Model | Linear Model | 0.024(-0.001, 0.050) | 0.016(-0.011, 0.043) | 0.013(-0.014, 0.041) |
| DenseNet121 | Linear Model | End-to-End | -0.001(-0.023, 0.019) | -0.016(-0.035, 0.001) | -0.027(-0.053, -0.003) |

Table 3:  AUC improvements achieved by MoCo-CXR-pretrained models against ImageNet-pretrained models on the Shenzhen tuberculosis task.

**AUPRC Performance of MoCo-CXR**   For all following figures, the green line represents MoCo-CXR pretrained models, whereas the blue line represents baseline Imagenet-pretrained models. Figures on the left compare models trained end-to-end and figures on the right compare performance of the corresponding linear models.

The first four images are AUPRC performance for the Pleural Effusion task from the CheXpert dataset. Here, we again observed that linear models based on MoCo-CXR pretrained representations consistently outperform those based on ImageNet-pretrained representations. In contrast, performance gains for end-to-end trained models are less significant.

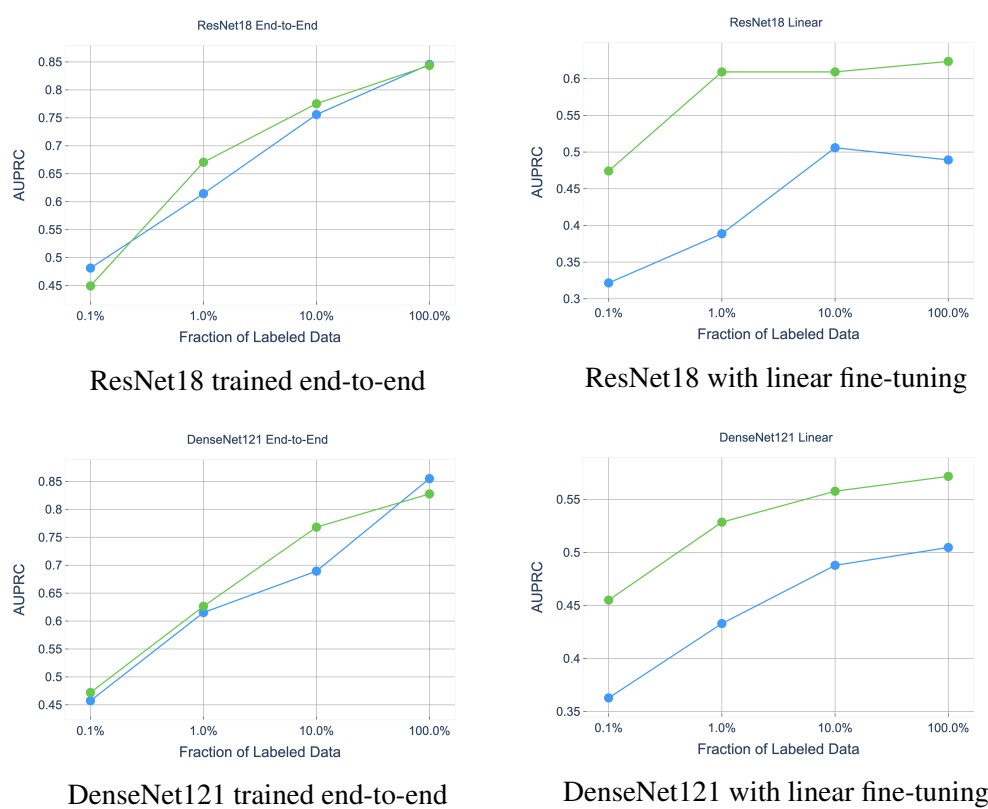

ResNet18 trained end-to-end                    ResNet18 with linear fine-tuning

DenseNet121 trained end-to-end               DenseNet121 with linear fine-tuning

Figure 3: Comparison of AUPRC performances for ResNet18-based and DenseNet121-based models on the Pleural Effusion task from the CheXpert dataset.

AURPC performance gains as visualized below for the Tuberculosis task on the Shenzhen dataset is roughly consistent with those observed on CheXpert and in line with AUROC performance discussed in Section 4.2.

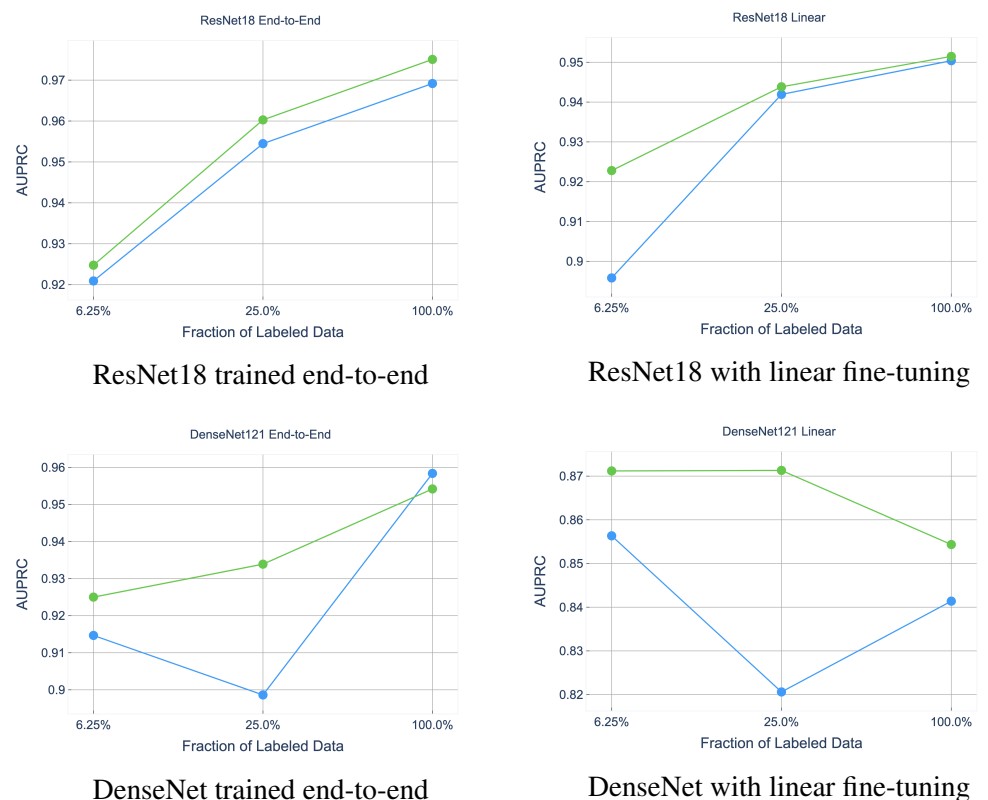

ResNet18 trained end-to-end · ResNet18 with linear fine-tuning

DenseNet trained end-to-end · DenseNet with linear fine-tuning

Figure 4: Comparison of AUPRC performances for ResNet18-based and DenseNet121-based models on the tuberculosis task from the Shenzhen dataset.

## Appendix C. MoCo-CXR Performance on Other CheXpert Tasks

To reinforce our results for pleural effusion as presented in the main paper, we also conducted follow-up experiments evaluating MoCo-CXR performance on the CheXpert competition task pathologies. These experiments were done exclusively with the ResNet18 backbone.

MoCo-CXR pretrained models outperforms ImageNet-pretrained models in linear evaluation on all five CheXpert competition tasks (Table 4). For these tasks, the most significant performance gain is observed at 0.1% label fraction and gains diminish with increasing label fraction, the same behavior as observed for the Pleural Effusion task. We also observe statistically significant performance gains with MoCo-CXR in end-to-end fine-tuning on all the CheXpert competition tasks (Table 5). This was observed at most label fractions for each pathology.

| | Baseline | MoCo-CXR | Improvement | Label Fraction |
|---|---|---|---|---|
| Atelectasis | 0.602(0.571, 0.635) | 0.630(0.592, 0.668) | 0.030(-0.005, 0.064) | 0.1% |
| | 0.612(0.574, 0.650) | 0.671(0.623, 0.714) | 0.060(0.022, 0.098) | 1% |
| | 0.703(0.654, 0.748) | 0.751(0.708, 0.796) | 0.048(0.011, 0.086) | 10% |
| | 0.732(0.685, 0.778) | 0.758(0.712, 0.802) | 0.025(-0.015, 0.065) | 100% |
| Cardiomegaly | 0.485(0.454, 0.516) | 0.640(0.605, 0.671) | 0.156(0.122, 0.193) | 0.1% |
| | 0.634(0.587, 0.674) | 0.735(0.690, 0.779) | 0.100(0.058, 0.139) | 1% |
| | 0.685(0.638, 0.733) | 0.745(0.698, 0.788) | 0.060(0.019, 0.101) | 10% |
| | 0.700(0.652, 0.747) | 0.737(0.694, 0.780) | 0.038(-0.004, 0.082) | 100% |
| Consolidation | 0.633(0.566, 0.694) | 0.644(0.554, 0.734) | 0.008(-0.085, 0.093) | 0.1% |
| | 0.615(0.548, 0.678) | 0.699(0.617, 0.771) | 0.083(-0.007, 0.166) | 1% |
| | 0.752(0.674, 0.821) | 0.794(0.699, 0.876) | 0.041(-0.059, 0.127) | 10% |
| | 0.749(0.665, 0.830) | 0.771(0.665, 0.859) | 0.019(-0.093, 0.119) | 100% |
| Edema | 0.725(0.682, 0.766) | 0.781(0.743, 0.814) | 0.055(0.016, 0.092) | 0.1% |
| | 0.810(0.764, 0.849) | 0.847(0.809, 0.883) | 0.038(0.005, 0.071) | 1% |
| | 0.844(0.801, 0.884) | 0.870(0.833, 0.900) | 0.024(-0.009, 0.059) | 10% |
| | 0.846(0.805, 0.886) | 0.867(0.830, 0.898) | 0.020(-0.015, 0.052) | 100% |
| Pleural Effusion | 0.683(0.644, 0.719) | 0.776(0.735, 0.811) | 0.096(0.061, 0.130) | 0.1% |
| | 0.766(0.716, 0.805) | 0.838(0.797, 0.873) | 0.070(0.029, 0.112) | 1% |
| | 0.799(0.750, 0.840) | 0.848(0.807, 0.883) | 0.049(0.005, 0.094) | 10% |
| | 0.815(0.768, 0.853) | 0.850(0.808, 0.884) | 0.034(-0.009, 0.080) | 100% |

Table 4: AUC improvements achieved by MoCo-CXR-pretrained linear models against ImageNet-pretrained linear models on CheXpert competition tasks

| | Baseline | MoCo-CXR | Improvement | Label Fraction |
|---|---|---|---|---|
| Atelectasis | 0.611(0.582, 0.641) | 0.673(0.637, 0.706) | 0.061(0.034, 0.089) | 0.1% |
| | 0.685(0.651, 0.719) | 0.730(0.693, 0.762) | 0.043(0.015, 0.069) | 1% |
| | 0.732(0.694, 0.770) | 0.787(0.750, 0.823) | 0.054(0.030, 0.076) | 10% |
| | 0.842(0.807, 0.878) | 0.821(0.781, 0.860) | -0.021(-0.037, -0.004) | 100% |
| Cardiomegaly | 0.593(0.561, 0.624) | 0.663(0.628, 0.695) | 0.069(0.041, 0.097) | 0.1% |
| | 0.728(0.690, 0.765) | 0.811(0.777, 0.840) | 0.083(0.059, 0.108) | 1% |
| | 0.808(0.769, 0.844) | 0.820(0.779, 0.856) | 0.011(-0.01, 0.029) | 10% |
| | 0.857(0.822, 0.890) | 0.858(0.824, 0.892) | 0.001(-0.014, 0.016) | 100% |
| Consolidation | 0.618(0.554, 0.680) | 0.646(0.574, 0.713) | 0.028(-0.015, 0.072) | 0.1% |
| | 0.673(0.614, 0.727) | 0.733(0.663, 0.792) | 0.061(0.006, 0.120) | 1% |
| | 0.809(0.763, 0.853) | 0.852(0.790, 0.903) | 0.042(0.009, 0.070) | 10% |
| | 0.888(0.847, 0.927) | 0.904(0.858, 0.939) | 0.016(-0.011, 0.046) | 100% |
| Edema | 0.771(0.737, 0.804) | 0.786(0.752, 0.820) | 0.015(-0.014, 0.042) | 0.1% |
| | 0.846(0.810, 0.880) | 0.850(0.815, 0.882) | 0.004(-0.017, 0.024) | 1% |
| | 0.865(0.829, 0.898) | 0.877(0.840, 0.908) | 0.011(-0.008, 0.030) | 10% |
| | 0.907(0.876, 0.935) | 0.894(0.860, 0.923) | -0.013(-0.027, 0.001) | 100% |
| Pleural Effusion | 0.775(0.737, 0.807) | 0.813(0.779, 0.842) | 0.037(0.015, 0.062) | 0.1% |
| | 0.858(0.856, 0.886) | 0.885(0.856, 0.909) | 0.027(0.006, 0.047) | 1% |
| | 0.898(0.872, 0.921) | 0.915(0.885, 0.937) | 0.017(0.003, 0.031) | 10% |
| | 0.942(0.920, 0.961) | 0.942(0.921, 0.921) | 0.000(-0.009 0.009) | 100% |

Table 5: AUC improvements achieved by MoCo-CXR-pretrained end-to-end models against ImageNet-pretrained end-to-end models on CheXpert competition tasks

