# OpenReview forum: "MoCo Pretraining Improves Representation and Transferability of Chest X-ray Models"
_MIDL.io/2021/Conference — MIDL 2021_

### Official Review · AnonReviewer4 · 2021-03-08

**Confidence:** 4
**Preliminary Rating:** 2
**Final Rating:** 2

**Summary:**

This paper shows an application of MoCo (Momentum Contrast) unsupervised learning on Chest X-ray classification tasks, called MoCo-CXR.
The difference between the MoCo and MoCo-CXR is different kinds of data augmentation techniques.
The authors evaluate their method on the CheXpert dataset and one external dataset from Shenzhen.


**Strengths:**

This paper shows the potential of unsupervised learning technicals like MoCo can also be beneficial on Chest X-ray classification tasks under different kinds of data augmentation ways.


.......................................

**Weaknesses:**

1. This paper states that some data augmentation methods are useful, while others not. I would like to see some result comparisons between different data augmentation methods. What's more, it is not clear which data augmentation methods are used, besides random rotation and horizontal flipping.
2. Some sentences are confusing to understand. For example, in Abstract "While contrastive learning has demonstrated promising results ... its application to medical imaging tasks ... has been limited". I think maybe the authors want to show that "its applications have not been well explored"?
3. This paper is about representation learning, but without any comparison with other representation learning methods, like SimCLR, PIPL.
4. What does label fraction mean in CheXpert training? Does it mean you use all the data to train but only 0.1\% (for example) data with labels?
5. From result Table 1 in the appendix, why Resnet18 has the same performance on both MoCo and ImageNet pretraining methods when using 100\% labeled data? Does it mean that unsupervised learning has no effect when with full supervision?
6. The improvement of MoCo compared to Imagenet is too small when using the end-to-end fine-tuning in Table 2.
7. Section 3.3 and 3.4 should be put into the Experiments section.
8. typos and some words not consistent. Like Moco-CXR, Shenzen dataset.

**Deanonymize Review:**

no

**Final Rating Justification:**

This paper is not self-contained and lacks comparison with other unsupervised learning methods for a technical paper.

**Justification Of The Preliminary Rating:**

It is an application of the MoCO method. However, the evaluation is not sufficient.


.....................................................................................................................

**Paper Type:**

validation/application paper

**Questions To Address In The Rebuttal:**




**Special Issue:**

no

---

> ### Author Response · Authors · 2021-03-18
> **We would like to thank the reviewer for their constructive comments.**
>
> We thank the reviewer for his/her time. Our responses are as below.
>
> * Insufficient evaluation
>   * Upon request, we have performed linear evaluation studies on all the CheXpert competition tasks. We found that MoCo-CXR consistently achieved higher AUC than baseline models at all label fractions. Our finding is included in the Appendix and the 0.1% label fraction result is summarized below. We will perform and add results from end-to-end experiments upon publication.
>
> | Task| Baseline | MoCo-CXR (ResNet18)|
> | :-------------| :------:   | -----------: |
> | Cardiomegaly| 0.486| *0.589*|
> | Normal / No finding| 0.786| *0.829*|
> | Consolidation| 0.627| *0.636*|
> | Edema| 0.737| *0.823*|
> | Atelectasis| 0.587| *0.647*|
>
> * I would like to see some result comparisons between different data augmentation methods. What's more, it is not clear which data augmentation methods are used, besides random rotation and horizontal flipping.
>   * We also believe that data augmentation is an important component of chest X-ray studies. The only augmentations used in this study are random rotations (+/- 10 degrees) and horizontal flipping. These data augmentations are the same as those used for CheXpert baseline (Irvin et al., 2019) and CheXNet (Rajpurkar et al., 2017). In a study on the effect of data augmentation for chest X-ray images (Data Augmentation for Chest Pathologies Classification, Sirazitdinov et al.) found that contrast, Gaussian noise and blur negatively affected diagnosis performance, while only random rotation, horizontal flipping and brightness demonstrated performance gain. We believe our choice for data augmentation is in-line with existing studies and the most suitable for CheXpert dataset.
>
> * Wording changes
>   * We have changed the abstract text to “While contrastive learning has demonstrated promising results on natural images … has not been well explored”
>
> * No comparison with other representation learning methods, like SimCLR, PIPL.
>   * We agree with the reviewer that other contrastive learning networks are also worthy of further investigations. However, we made the decision to use MoCo given its substantially reduced dependency on the batch size, reducing requirement for computational resources and making all our experiments possible on a single GPU. In contrast, SimCLR (Chen et al.) draws negative samples from the training batches, requiring large batch sizes on the order of thousands of images; MoCo v2 (Chen et al., 2020c) found that SimCLR had far lower performance than MoCo with lower batch sizes. Our goal was to demonstrate that contrastive learning could be used for chest X-rays; comparing multiple frameworks was out of the scope of the current project.
>
> * What does label fraction mean in CheXpert training? Does it mean you use all the data to train but only 0.1% (for example) data with labels?
>   * Correct; the label fraction refers to the quantity of labeled data used in the fine-tuning step. For example, with 0.1% label fraction, all data are used for the unsupervised pretraining, but only 0.1% of the labels are used for fine-tuning in the semi-supervised learning setup. Labels for the remaining 99.9% are hidden from the model. The term “label fraction” was also used for SimCLR (Chen et al.)
>
> * From result Table 1 in the appendix, why Resnet18 has the same performance on both MoCo and ImageNet pretraining methods when using 100% labeled data?
>   * It is true that performance improvement decreases with larger label fractions, which is expected as the model without the pretrained-initialization is able to catch up with more labeled data. For example, SimCLR found that, for many tasks, non-pretrained models also achieved equal or near-equal performance to the self-supervised models with 100% labels. Nevertheless, the improvements we demonstrate at smaller label fractions have strong clinical relevance; in practice, we will want models with high sample efficiency that can quickly transfer to new pathologies that lack many labeled X-rays.
>
> * The improvement of MoCo compared to Imagenet is too small when using the end-to-end fine-tuning in Table 2.
>   * We believe the reviewer is referring to Supplementary Table 2 instead of Table 2 in the main text. Here, we agree that the improvement of end-to-end training is small on ResNet 18. However, the value demonstrated here is that representations learned using images from CheXpert can be transferred to the Shenzhen dataset, whereas ImageNet-pretrained representations are not transferable. For this purpose, the most important comparison is on performance of fine-tuning the last linear layer. For this purpose our MoCo-CXR demonstrated substantial performance gain at low label fraction on ResNet 18 compared to an equivalent ImageNet-pretrained model.
>
> * Other minor errors
>   * We thank the reviewer for pointing this out. We will thoroughly review our text and correct these minor errors at the time of publication.

---

### Official Review · AnonReviewer2 · 2021-03-08

**Confidence:** 5
**Preliminary Rating:** 1

**Summary:**

The paper demonstrates an application of a self-supervised method based on contrastive learning, to produce deep-learning models that provide patient representations, that are highly discriminative of pathologies in chest X-rays. The authors evaluated their proposed pipeline in the context of a large chest x-ray study.

**Strengths:**

The paper proposes a self-supervised pre-training pipeline to learn patient representations that are highly discriminative of some diseases. It's an interesting direction given, large unlabeled datasets, especially for a chest x-ray.

**Weaknesses:**

1. The paper lacks motivation and failed to justify the need for the proposed pre-training pipeline. In related work, the authors provide evidence that ImageNet initialized models, may not always optimally perform for medical images. While in the experiment section, the authors only compared against such ImageNet pre-trained models and also made Image-Net-based initialization a part of their pipeline. Given the related work and limited comparisons in the experiments, the scope and application of the proposed pipeline are not clear or justified.

2. The method section provided no details regarding the MoCo self-supervision method. The manuscript is not self-contained and required readers to refer to the related work to understand this paper.

3. Limited evaluation of the proposed method on large chest x-ray datasets. The authors considered only a single classification task on a balanced dataset. The experiments on large chest x-ray datasets are evaluated on only Pleural effusion. More pathologies should be considered to demonstrate the generalization of the learned representations.

4. Very often, the labeled dataset for medical images has a significant class-imbalance. An evaluation of how the proposed model performance on an imbalanced dataset is missing.

5.  The experiment section repeats the observations from the figures and tables and fails to discuss or explain the results in the context of the experiment.
Ony reporting AUC metric is not sufficient to demonstrate the superior performance of any method, especially in medical images.  The authors should consider some exploration analysis, like investigating the population on which their method was more discriminative than others, any interpretability benefits of using their representations, etc.

6. A linear model on top of representations learned from a self-supervised learning approach is interesting, as may provide some easier-to-understand insights into the networks' decision-making process. The paper does not provide any such reasoning.

7. Limited or no comparisons with other self-supervised methods for learning representations for medical images.

8. The proposed contrasting training strategy for selecting positive pairs has previously been used in the context of medical imaging [1].

[1] Sun, L., Yu, K. and Batmanghelich, K., 2020. Context Matters: Graph-based Self-supervised 	Representation Learning for Medical Images. arXiv preprint arXiv:2012.0645 AAAI-2020.



**Deanonymize Review:**

no

**Justification Of The Preliminary Rating:**

The paper proposes an interesting problem of using large unlabelled datasets to learn patient representations that are relevant for downstream tasks such as disease classification. But current manuscript fails to provide relevant method details and sufficient experiments to supports their proposed pipeline.

**Paper Type:**

validation/application paper

**Questions To Address In The Rebuttal:**

1. The authors should consider expanding their method section to provide a summarization of the contrastive learning approach.

2. The authors should consider expanding their experiment section to add results for more pathology classification tasks.

3. The authors should consider elaborating on their results beyond the ROC-AUC metric, to provide an insight into the significance of their learned representations.

**Special Issue:**

no

---

> ### Author Response · Authors · 2021-03-18
> **We would like to thank the reviewer for their constructive comments.**
>
> We thank the reviewer for his/her time. We have addressed concerns raised by the reviewer and modified our paper. Our responses are as below.
>
> * Limited or no comparisons with other self-supervised methods for learning representations for medical images.
>   * The primary goal of this work was to demonstrate that a contrastive learning framework could be used to develop more performant and sample-efficient chest X-ray models than the current pretraining (ImageNet) paradigm allows. Our goal was not to compare different contrastive learning techniques, even though this certainly represents an avenue for future work. We chose MoCo to demonstrate our hypothesis given its substantially reduced dependency on the batch size, reducing the requirement for computational resources and making all our experiments possible on a single GPU. SimCLR, for example, draws negative samples from the training batches, requiring large batch sizes on the order of thousands of images.
>
> * The proposed contrasting training strategy for selecting positive pairs has previously been used in the context of medical imaging.
>   * We would like to point out that this reference does not precede our work and focuses on CT scans rather than chest X-rays. The volumetric nature of CT scans requires a task-specific form of augmentation that would not directly transfer to chest X-rays. Indeed, in Related Work we referenced a paper (Chaitanya et al., 2020), which was published prior to the reviewer’s reference, that proposed a strategy to extract contrastive pairs from MRI scans utilizing their volumetric nature and discussed why such an approach would not apply to chest X-rays.
>
> * The authors should consider expanding their method section to provide a summarization of the contrastive learning approach.
>   * We agree with the reviewer that a summarization of the contrastive learning approach, and in particular the momentum contrastive learning approach, should be added to the methods section. We have added an illustration of contrastive learning to Section 2, another illustration of the momemtum contrast in the Appendix and an illustration of data augmentation in the Appendix.
>
>   * We will also added content similar to the following upon publication
>
>     * *Contrastive learning is a method of representation learning, in which positive encodings, those generated from transformations of the original image, are compared against encodings of other images. Contrastive loss, which is usually variants of InfoNCE, in turn penalizes proximity of positive-negative pairs in the lower-dimensional embedding space. Momentum contrast stands out from other contrastive learning methods because it uses a momentum weighted negative embedding, which reduces dependency on training batch size, therefore decreases hardware requirement for contrastive learning models.*
>
> * The authors should consider expanding their experiment section to add results for more pathology classification tasks.
>   * Upon request, we have performed linear evaluation studies on all the CheXpert competition tasks. We found that MoCo-CXR consistently achieved higher AUC than baseline models at all label fractions. Our finding is included in the Appendix and the 0.1% label fraction result is summarized below. We will perform and add results from end-to-end experiments upon publication.
>
> | Task                       | Baseline | MoCo-CXR (ResNet18)     |
> | :-------------             | :------:   | -----------: |
> | Cardiomegaly               | 0.486      | *0.589*        |
> | Normal / No finding               | 0.786      | *0.829*        |
> | Consolidation	         | 0.627      | *0.636*        |
> | Edema                      | 0.737      | *0.823*        |
> | Atelectasis                | 0.587      | *0.647*        |
>
>
>   * In addition to pathologies labeled for CheXpert, we assessed the transferability of the MoCo-CXR-pretrained representations to the Shenzhen dataset, which was unseen during any pretraining. The Shenzhen task is Tuberculosis, so we are able to simultaneously assess whether our method transfers to a new pathology as well as an external dataset.
>
> * The authors should consider elaborating on their results beyond the ROC-AUC metric, to provide an insight into the significance of their learned representations.
>   * Upon the reviewer’s request, we have added graphs/tables of our models’ AUPRC for the pleural effusion and TB tasks to the appendix. Results observed for AUPRC are inline with we have observed for AUC. We note that most current contrastive learning frameworks (MoCo, SimCLR, PIRL) use AUC or accuracy@top-K of a linear classifier trained on the pretrained representations as a proxy for representation quality. Given its ubiquity, we determined that this technique would make the most sense to measure the significance of the learned representations.

---

### Official Review · AnonReviewer3 · 2021-03-09

**Confidence:** 3
**Preliminary Rating:** 3
**Recommendation:** Oral, Poster

**Summary:**

They proposedMoCo-CXR, which is an adaptation of the contrastive learning method Momentum Contrast (MoCo), to produce models with better representations and initializations for the detection of pathologies in chest X-rays.
In results, they found that MoCo-CXR-pretraining provided the most benefit with limited labeled training data.

**Strengths:**

They customized a loss function with metadata-based positive pairs in the contrastive learning extracting more general features.
The negative pair is good enough to be adjusted for training the models by selecting them with consideration of metadata.

**Weaknesses:**

The lack of introduction to data split or dataset with different ratio of dataset. (Which part of dataset is used for validation and test)
More detailed hyperparameters used for tuning the models are needed such as the size of the input image, etc.



**Deanonymize Review:**

no

**Justification Of The Preliminary Rating:**

The authors tried to validate the effectiveness of contrastive learning with various experiments related to the real environment.

Additionally, they used the metadata of patients to sample datasets for training models efficiently, which concept could be adjusted in other tasks as well.

**Paper Type:**

validation/application paper

**Questions To Address In The Rebuttal:**

1. The lack of introduction to data split or dataset with different ratio of dataset.
Can you add more information on the dataset construction?

2. More detailed hyperparameters used for tuning the models are needed such as the size of the input image, etc.
Could you put more details?

3. Have you ever tested on another dataset?

**Special Issue:**

no

---

> ### Author Response · Authors · 2021-03-18
> **We thank the reviewer for their encouraging and constructive comments.**
>
> We thank the reviewer for his/her time. We have addressed concerns raised by the reviewer and modified our paper. Our responses are as below.
>
> * The lack of introduction to data split or dataset with different ratio of dataset. Can you add more information on the dataset construction?
>   * We agree with the reviewer that a brief introduction to splitting data for the purpose of semi-supervised learning is needed. We have now added the following content to Section 3.3.
>
>   * We use label fraction to represent the ratio of data with its labels retained during training. For a model trained with 1% label fraction, the model will only have access to 1% of the all labels, while the remaining 99% of labels are hidden from the model. The use of label fraction is a proxy for the real world, where large amounts of data remain unlabelled and only a small portion of well-labelled data can be used toward supervised training.*
>
> * More detailed hyperparameters used for tuning the models are needed such as the size of the input image, etc. Could you put more details?
>   * We agree with the reviewer that more details for the datasets used can be included. It is worth noting that regardless of the size of original images, all inputs are resized to 224 x 224 to stay consistent with CheXpert implementation. We have added the following content to Section 3.1.
>   **Chest X-ray images included in the CheXpert dataset are of size 320 x 320*. This fact is also mentioned in Section 3.2. Then *Chest X-ray images included in the Shenzhen dataset are of size 4020 x 4892 and 4892 x 4020.*
>
> * Have you ever tested on another dataset?
>   * Yes, we evaluated whether the learned representations from CheXpert transferred to the Shenzhen dataset, which was unseen during pretraining. The Shenzhen task is Tuberculosis, so we are able to simultaneously assess whether our method transfers to a new pathology as well as an external dataset.
>   * We have also added new results from linear evaluation studies on all the CheXpert competition tasks in order to assess our method's performance on multiple pathologies. We found that MoCo-CXR consistently achieved higher AUC than baseline models at all label fractions. Our finding is included in the Appendix and the 0.1% label fraction result is summarized below. We will perform and add results from end-to-end experiments upon publication.
>
> | Task                       | Baseline | MoCo-CXR (ResNet18)     |
> | :-------------             | :------:   | -----------: |
> | Cardiomegaly               | 0.486      | *0.589*        |
> | Normal / No finding               | 0.786      | *0.829*        |
> | Consolidation	         | 0.627      | *0.636*        |
> | Edema                      | 0.737      | *0.823*        |
> | Atelectasis                | 0.587      | *0.647*        |

---

### Official Review · ~Pedro_M._Gordaliza1 · 2021-03-09

**Confidence:** 4
**Preliminary Rating:** 3
**Recommendation:** Poster

**Summary:**

The authors adapt to the field of medical imaging, MoCo, a framework based on Contrastive Learning that despite its unfortunate name for Spanish speakers :-), has shown a high capacity to produce good results in classification problems of different kinds in Computer Vision.
Applying MoCo to the specific problem of X-Ray image classification, the authors propose a new variant to alleviate a classic problem in most medical imaging scenarios: the lack of labelled data.
To test the feasibility of adapting MoCo to X-ray (MoCo-CXR), first, a model is pre-trained under the MoCo framework, then it is used as initialization for different models that aim to classify X-ray images as images where the manifestation pleural effusion appears. To evaluate the performance under this initialization, experiments are performed mixing: a) different architectures (ResNet18 and DenseNet121), b) training a classification model "on top" of the architecture or end-to-end, c) using different percentages of labelled data (0.1%, 1%, 10% and 100%) and d) with MoCo pretraining or with Imagenet pretraining.

**Strengths:**

- In general, the paper is well written
- A computer Vision SOTA method is adapted to the Medical imaging field.
- The statistical analysis is clear
- Well evaluated methods to handle the lack of data in Medical Imaging are always welcome.
- Results are promising

**Weaknesses:**

- The concept of Contrastive Learning is not very clear. It is likely that many of the MIDL audience is not familiar with it.
- Code not available
- Other contrast learning approaches are mentioned, including in Medical Imaging, but no comparison is made.
- No mention is made of the limitations of obtaining CIs of bootstrap estimators.
- There is not a single image in the paper to illustrate concepts as pleural effusion, the different augmentations, etc.
- That classification results are only given for pleural effusion, in a dataset such as CheXpert that contains many more labels, is very striking in terms of generalization of the method.

**Deanonymize Review:**

yes

**Justification Of The Preliminary Rating:**

The authors have shown that MoCo can be a good option to alleviate the lack of labels in classification problems from X-ray images or at least an option worth studying. Unfortunately, the validation suffers from some generalisation problems. Still, it is an interesting enough work to be presented at MIDL2021.

**Paper Type:**

validation/application paper

**Questions To Address In The Rebuttal:**

- It is necessary to explain contrastive learning concisely to the general audience.

- Figure 1, mixing datasets, pre-entrant architectures and final classification models is very ambiguous for a reader who wants to grasp the idea at a glance. Please, could you review it?

- The charts in figure 4 are not properly identified

- The experiments carried out make it clear that there is an improvement in results when using MoCo pretraining but it would be interesting to see comparisons with other methods to evaluate to what extent the computational efficiency is worthwhile:
a) comparison with others Contrastive learning framewoks. SimCLR could work with smaller batches if training longer, right?
b)  comparison with other SSL methods. i.e: pseudo-labelling. The differences with the Imagenet pretraining is not that big, especially in the TB dataset (differences are not significant) and it would be really interesting to compare the performance with another approach
c) Taking into account the performance difference aforementioned, could we compensated with a most dense model?

- The proportions of labelled data employed for CheXpert and TB dataset are quite different, why?

- There are many papers on the possible "correct" transformations that can be made to X-ray images, why have you focused on only two types? How would it change the performance to include more?
In the case of trying to generalise MoCo for any radiological image (e.g. CT, MRI, PET), which transformations could always be used?





**Special Issue:**

no

---

> ### Author Response · Authors · 2021-03-18
> **We thank the reviewer for their encouraging and constructive comments.**
>
> We have included point-by-point responses to the reviewer's questions below.
>
> * The code repository is in preparation and it will be released on Github upon publication.
>
> * *Limitations of Boostrapping*: We utilized the bootstrap on the test set to determine variability (i.e CIs) in the performance of the estimators over the test set. Our estimators were produced via fine-tuning on specific label fractions of CheXpert/Shenzhen; we did not use a method like bagging to bootstrap the training set.
>
> * *Additional figures*: We have included additional images in the methods section to illustrate the process of contrastive learning. In addition, we have also added illustrations for momemtum contrast and examples of data augmentation (using an example of Pleual Effusion) in the Appendix section.
>
> * *Additional evaluations*: We assess the generalization of our method by evaluating whether MoCo-CXR-pretrained representations transfer to the Shenzhen dataset, which was unseen during any pretraining. The Shenzhen task is Tuberculosis, so we are able to simultaneously assess whether our method transfers to a new pathology as well as an external dataset.
>   * Upon request, we have performed linear evaluation studies on all the CheXpert competition tasks. We found that MoCo-CXR consistently achieved higher AUC than baseline models at all label fractions. Our finding is included in the Appendix and the 0.1% label fraction result is summarized below. We will perform and add results from end-to-end experiments upon publication.
>
> | Task                       | Baseline | MoCo-CXR (ResNet18)     |
> | :-------------             | :------:   | -----------: |
> | Cardiomegaly               | 0.486      | *0.589*        |
> | Normal / No finding               | 0.786      | *0.829*        |
> | Consolidation	         | 0.627      | *0.636*        |
> | Edema                      | 0.737      | *0.823*        |
> | Atelectasis                | 0.587      | *0.647*        |
>
>
> * We will add description of contrastive learning to the methods section. Currently, we have included illustrations for contrastive learning and clarified on the process of MoCo-CXR.
>
> * *Figure 1*: We have modified the previous Figure 1 to show the 2 flows (CheXpert and Shenzhen) in parallel in hope to increase legibility of this figure.
>
> * *Figure 4*: We have modified the figure caption to “AUC on the Shenzhen tuberculosis tasks shows that MoCo-CXR pretraining still introduces significant improvements despite being fine-tuned on the CheXpert dataset. The top row compares AUC of MoCo-CXR models with a trained linear classifier. The bottom row compares AUC of MoCo-CXR models that were trained end-to-end.”
>
> * *Comparison with other SSL*: The primary goal of this work was to demonstrate that a contrastive learning framework could be used to develop more performant and sample-efficient chest X-ray models than the current pretraining (ImageNet) paradigm allows. Comparing other contrastive learning and SSL frameworks would be an excellent opportunity for future work. For this project, we chose MoCo over a framework like SimCLR because of its reduced dependency on the batch size; furthermore, the MoCo authors found that their method significantly outperformed SimCLR under the same epochs and batch size, and also outperformed SimCLR even when the SimCLR model was trained for 200 epochs longer (Table 2 of Chen et. al., 2020).
>
> * *Proportion of labeled data*: The conventional split for ImageNet-based semi-supervised learning studies typically use 1%, 10% and 100% label fraction, which we followed for the larger CheXpert dataset (100k+ images). However, the Shenzhen dataset is much smaller, with only 600+ images in total. Having a 1% label fraction would mean the training sample would be less than 6, making the semi-supervised learning model heavily dependent on randomization of training sample selection. Using a ¼ (25%) and 1/16 (6.25%) ensures an exponential split while maintaining a reasonable number of samples (approximately 25 images) at the lowest label fraction. To address the issue associated with randomization, we performed multiple runs for these lower label fractions and performance of these results are averaged to ensure stability, as explained in Section 3.2.
>
> * *Data transformation*: The purpose of our work was not to compare and contrast the different transformation methods, but use data augmentations commonly used in the chest X-ray setting. In this case, the two augmentations were the same as used for the CheXpert baseline models in Irvin et al., 2019.

---

### Meta-Review · Area_Chairs · 2021-03-28

**Recommendation:** Accept (Poster)

**Metareview:**

2/4 Reviewers were very supportive of the paper whilst 2/4 raised strong concerns in terms of the level of clarity of the technique and experiments.  Despite R2 and R4 have some level of concerns on the technique and the experimental setting, they also recognise some strengths of the paper -- particularly, in the challenge of handling a vast amount of unlabelled medical data.   The AC reads the paper, rebuttal and discussions. The AC weights the strong points of each reviewer and the responses from the authors, and as a result, recommend Accept (poster).

**Paper Type:**

validation/application paper

---

### Decision · Program_Chairs · 2021-03-31

Accept